# Physically Embodied Gaussian Splatting: A Visually Learnt and Physically Grounded 3D Representation for Robotics

**Jad Abou-Chakra**
Queensland University of Technology

**Krishan Rana**
Queensland University of Technology

**Feras Dayoub**
The University of Adelaide

**Niko Sünderhauf**
Queensland University of Technology

**Abstract:** For robots to robustly understand and interact with the physical world, it is highly beneficial to have a comprehensive representation – modelling geometry, physics, and visual observations – that informs perception, planning, and control algorithms. We propose a novel dual "Gaussian-Particle" representation that models the physical world while (i) enabling predictive simulation of future states and (ii) allowing online correction from visual observations in a dynamic world. Our representation comprises particles that capture the geometrical aspect of objects in the world and can be used alongside a particle-based physics system to anticipate physically plausible future states. Attached to these particles are 3D Gaussians that render images from any viewpoint through a splatting process thus capturing the *visual* state. By comparing the predicted and observed images, our approach generates "visual forces" that correct the particle positions while respecting known physical constraints. By integrating predictive physical modeling with continuous visually-derived corrections, our unified representation reasons about the present and future while synchronizing with reality. We validate our approach on 2D and 3D tracking tasks as well as photometric reconstruction quality. Videos are found at https://embodied-gaussians.github.io/.

## 1 INTRODUCTION

The real world is governed by many well-understood physical priors – matter cannot occupy the same space, scenes comprise rigid and deformable objects, gravity acts on all objects, robots have known kinematic structures. Incorporating such priors into a world representation can constrain how it evolves over time, enforcing adherence to the laws of physics. However, most representations like pointclouds, images [1], or latent descriptors [2, 3, 4] cannot explicitly encode and reason over these priors. Consequently, their ability to predict future states lacks critical physical constraints.

Particle-based physics simulators [5, 6, 7] elegantly capture the physical priors that are often known in robotic scenarios and enable forward simulation of dynamics. This makes them attractive for modeling the physical world. To use them in a robotics context, we propose a method to initialize particles from RBGD observations and to periodically correct the errors accrued over time using only RGB observations from the real world.

To enable continuous state correction through visual feedback, we add a visual aspect to the particles which allows a corrective force to be calculated. Recent work has shown 3D Gaussians [8, 9] are a differentiable, performant, and expressive representation of visual state that can render images from any viewpoint. Our contribution is to couple these Gaussians to the particles. With this dual Gaussian-Particle representation, we can simulate future states using a physics system. We can also predict the visual appearance by rendering the attached Gaussians. Observations are compared to the rendered images to compute a photometric loss which drives the movement of the Gaussians and subsequently generates "visual forces" that correct the positions of the attached particles. Our key contributions are thus: (i) A novel dual Gaussian-Particle representation that captures geometry, physics, and visual appearance in a unified way. (ii) A method of initializing this representation from RGBD data and instance maps. (iii) A real-time method to correct the particle states using visual feedback. An overview of our system is shown in Fig. 1.

8th Conference on Robot Learning (CoRL 2024), Munich, Germany.

Figure 1: Image from a real-world experiment showing the available physical priors (1), the dual Gaussian-Particle representation (2), the predicted visual state of the world (3), and the corrective "visual" forces (5) being applied as a result of a visual discrepancy between the rendered state and the image from the camera (4).

## 2 PRELIMINARIES

Our approach builds upon Position-Based Dynamics [5, 6] (PBD) and Gaussian Splatting [8]. This section provides an overview of these two components separately, while the subsequent section details our contribution that interconnects them for a robotics setting.

**Particle-Based Physics Simulation** PBD is a physics simulation technique well-suited for our robotics application, which requires real-time operation and robust performance. In our formulation, PBD acts on oriented particles where each particle $i$ is defined by its position $\boldsymbol{p}_i \in \mathbb{R}^3$, velocity $\boldsymbol{v}_i \in \mathbb{R}^3$, orientation $\boldsymbol{q}_i \in \mathbb{S}^3$, angular velocity $\boldsymbol{\omega}_i \in \mathbb{R}^3$, external force $\boldsymbol{f}_i \in \mathbb{R}^3$, radius $r_i \in \mathbb{R}^+$, and mass $m_i \in \mathbb{R}^+$. A particle may belong to a shape $S_j$ and thus has its resting position $\bar{\boldsymbol{p}}_i$ as an additional attribute.

At the core of the PBD framework are the various constraints that govern the behavior of the simulation. These constraints are defined as cost functions that operate on the particle positions, ensuring the simulation adheres to the desired physical properties. The general form of a PBD constraint is a cost function $C(\mathbf{p}_0, ..., \mathbf{p}_n, \mu_r) \in \mathbb{R}^+$, where $\mu_r$ is a relaxation factor. The constraints are minimized at each simulation step using a Jacobi solver. The solver iteratively updates the positions of the particles by aggregating the required positional changes $\Delta \boldsymbol{p}$ to locally satisfy the constraint. Generally, the change required is given by:

$$\Delta \boldsymbol{p}_i = -\mu_r w_i \frac{C}{\sum_j w_j |\nabla_{\boldsymbol{p}_j} C|^2} \cdot \nabla_{\boldsymbol{p}_i} C \quad \text{where} \quad w_i = \frac{1}{m_i} \tag{1}$$

In our system, we use ground, collision, and shape constraints. Their associated $\Delta \boldsymbol{p}$ are outlined below:

PBD's **ground constraint** is used to prevent particles from penetrating the ground plane given by $(\boldsymbol{n}, d)$:

$$\Delta \boldsymbol{p}_i^{\text{ground}} = C_{\text{ground}}(\boldsymbol{p}_i; \boldsymbol{n}, d) \cdot \boldsymbol{n}, \quad C_{\text{ground}}(\boldsymbol{p}_i; \boldsymbol{n}, d) = \min(\boldsymbol{n}^T \boldsymbol{p}_i + d - r_i, 0) \tag{2}$$

PBD's **collision constraint** which operates on a pair of particles $i$ and $j$ is used to model collisions:

$$\Delta\boldsymbol{p}_i^{\text{col}} = \frac{w_i}{w_i + w_j} \frac{\boldsymbol{p}_i - \boldsymbol{p}_j}{||\boldsymbol{p}_i - \boldsymbol{p}_j||} C_{\text{col}(\boldsymbol{p}_i, \boldsymbol{p}_j)}, \quad C_{\text{col}}(\boldsymbol{p}_i, \boldsymbol{p}_j) = \min(||\boldsymbol{p}_i - \boldsymbol{p}_j|| - r_i - r_j, 0) \quad (3)$$

Lastly, to ensure a group of particles belonging to a particular object (either deformable or rigid) maintain their structure throughout the simulation, we use the **shape matching** algorithm which is explained in detail in [7, 10]. Briefly, the shape matching algorithm requires the computation of the following matrix for each shape $S$:

$$\boldsymbol{A}_S = \sum_{i \in S} \frac{1}{5} m_i \boldsymbol{R}_i + \boldsymbol{p}_i \bar{\boldsymbol{p}}_i^T - M \boldsymbol{c}_S \bar{\boldsymbol{c}}_S^T, \quad \boldsymbol{c}_S = \frac{\sum_{i \in S} m_i \boldsymbol{p}_i}{M}, \quad \bar{\boldsymbol{c}}_S = \frac{\sum_{i \in S} m_i \bar{\boldsymbol{p}}_i}{M}, \quad M = \sum_{i \in S} m_i \quad (4)$$

where $\boldsymbol{R}_i \in \mathbf{SO(3)}$ is the matrix form the quaternion $\boldsymbol{q}_i$. $\boldsymbol{A}_S$ can be decomposed into $\boldsymbol{R}_S \boldsymbol{S}$ and thus the changes in particles positions required to maintain structure are given by:

$$\Delta\boldsymbol{p}_i^{\text{shape}} = k_S[\boldsymbol{R}_S(\bar{\boldsymbol{p}}_i - \bar{\boldsymbol{c}}_S) + \boldsymbol{c} - \boldsymbol{p}_i] \quad (5)$$

Here, $k_S$ is the stiffness parameter of the shape. Both rigid and deformable objects can be modelled with shape matching. A rigid object is composed of a single shape. A deformable object, however, is composed of multiple shapes where each shape is composed of a particle and its neighbours.

For a sample implementation of PBD and the shape matching algorithm, we refer the reader to [11]. In our work, we build on Warp's [12] PBD implementation and extend it to incorporate the shape matching algorithm. In our experiments, each physics step takes approximately 5 ms to complete. More details on PBD can be found in our supplementary.

**Gaussian Splatting**  Gaussian splatting [8] has emerged as a powerful rendering technique that can capture the state of the visual world with a discrete set of 3D Gaussians. Each Gaussian $i$ is parameterized by its position $\boldsymbol{g}_i \in \mathbb{R}^3$, orientation $\boldsymbol{R}_i \in \mathbf{SO(3)}$, scale $\boldsymbol{s}_i \in \mathbb{R}^3$, opacity $\alpha_i \in \mathbb{R}^+$, and color $\boldsymbol{c}_i \in \mathbb{R}^3$.

Given a viewpoint whose transform relative to the world frame is denoted by $V \in \mathbf{SE(3)}$ and projection function from the 3D world to the view's screenspace is defined by $\pi(\mathbf{x})$, the color at a pixel coordinate $\mathbf{u}$ can be calculated by sorting the Gaussians in increasing order of their viewspace z-coordinate and then using the splatting formula in Eq. (6).

$$C_{\text{rgb}}(\mathbf{u}) = \sum_{i \in \mathcal{N}} c_i \alpha_i(\mathbf{u}) \prod_{j=1}^{i-1} (1 - \alpha_j(\mathbf{u})) \quad (6)$$

$$\alpha_i(\mathbf{u}) = a_i e^{-g_i(\mathbf{u})}, \quad g_i(\mathbf{u}) = \mathbf{x}_i^T \boldsymbol{\Sigma}_i'^{-1} \mathbf{x}_i, \quad \mathbf{x}_i = \mathbf{u} - \pi(\boldsymbol{g}_i)$$

$\boldsymbol{\Sigma}_i' = \mathbf{J} \mathbf{V} \boldsymbol{\Sigma}_i \mathbf{V}^T \mathbf{J}^T$ is the covariance of the Gaussian $i$ projected into the viewpoint's screenspace where $\mathbf{J}$ is the Jacobian of the projection function $\pi(\mathbf{x})$ and $\boldsymbol{\Sigma}_i = \boldsymbol{R}_i \text{diag}(s_i^2) \boldsymbol{R}_i^T$. For the full details of this process, the reader is referred to [8].

The rendering equation is not limited to color. In our method, we also associate each Gaussian with a segmentation id $o_i$ (as is done in [9]). Note, however, this is only needed for our initialization scheme and not our prediction and corrective steps. Rendering segmentation is done using:

$$S(\mathbf{u}) = \sum_{i \in \mathcal{N}} o_i \alpha_i(\mathbf{u}) \prod_{j=1}^{i-1} (1 - \alpha_j(\mathbf{u})) \quad (7)$$

Since the splatting process is differentiable, the attributes defining the 3D Gaussians can be learnt to represent a specific scene by minimizing the photometric loss $L_{\text{rgb}}$ between a set of groundtruth images and their corresponding splatted renders. Our initialization procedure also makes use of $L_{\text{seg}}$.

$$L_{\text{rgb}} = \sum_{\boldsymbol{u}} |C_{\text{rgb}}(\mathbf{u}) - C_{\text{gt}}(\mathbf{u})| \quad \text{and} \quad L_{\text{seg}} = \sum_{\boldsymbol{u}} |S(\mathbf{u}) - S_{\text{gt}}(\mathbf{u})| \quad (8)$$

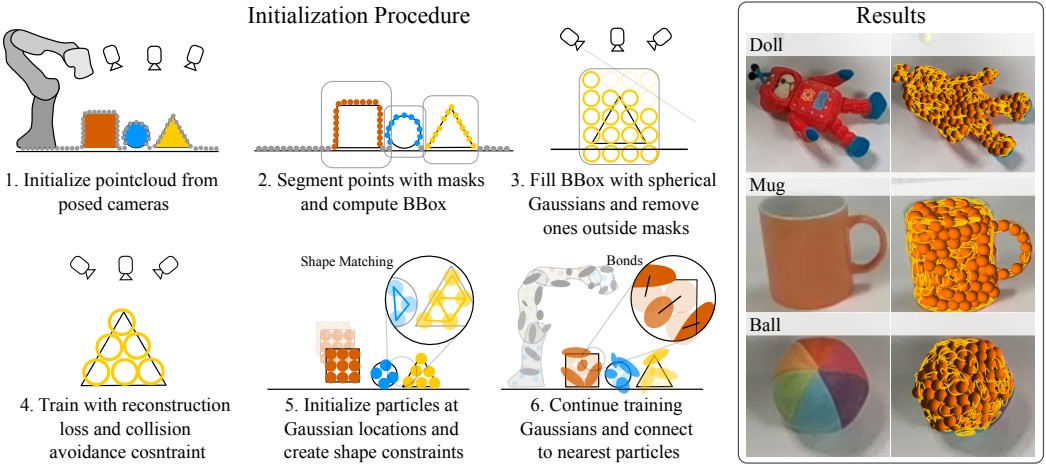

Figure 2: The initialization procedure (left) and example results from real-world data (right).

# 3 METHOD

Our method creates a model of the world that can at realtime rates be (i) forward simulated, (ii) regularized with physical priors, and (iii) corrected through visual observations from 3 cameras. The model comprises two tightly integrated representations: a set of $N$ particles that represent the physical world and are acted upon by a PBD physics system, and $M$ Gaussians that visually depict the world through efficient splatting-based rendering. The key novelty lies in the introduction of "Gaussian-Particle" bonds that synergistically couple these two representations, creating a bridge between the physical and visual aspects of the modeled environment. A bond is a rigid transform that links a Gaussian to its parent particle. In Sec. 3.1, we describe how to initialize the particles, Gaussians, and their interconnecting bonds. In Sec. 3.2, we describe how the simulated representation is kept synchronized with the real world using the observations from the cameras.

## 3.1 Initialization

To initialize the particles, we compute a loose 3D bounding box around each object using the depth data and instance masks. The handful of instance masks required are user-generated with Cutie [13] (a mask labelling tool) in only a few seconds with minimal effort, however this could plausibly be automated using VLMs or SAM [14, 15, 16]. The instance and depth information is only required at initialization. The bounding boxes are filled with evenly spaced spherical Gaussians whose radius matches the smallest relevant geometric feature (4 to 7 mm). Gaussians that do not project into the instance mask are pruned. The Gaussian positions, colors, and opacities are optimized by iteratively solving for collision and ground constraints using the Jacobi solver mentioned in (Sec. 2) and minimizing the photometric and segmentation reconstruction loss (Eq. (8)) using Adam [17]. Thus for this stage only, the Gaussians act as though they are also particles. Gaussians with an opacity lower than 0.3 (emperically chosen) are pruned and their size is upper bounded to roughly the size of the particle. Particles are initialized at the locations of the remaining Gaussians. The particles belonging to each object are then connected to each other using shape matching constraints (2).

The user also indicates whether each object is rigid or deformable. We envision that in the future, a VLM could automatically make this determination. These particles represent a collision-free, ground-aligned approximation of the object geometries, filling up the observed shapes. While this approach does not model cavities, this could be addressed in the future by incorporating depth-based losses or by replacing the grid initialization with a more sophisticated 3D initialization scheme.

Subsequently, we continue optimizing the Gaussians without imposing collision constraints and allow the scale to change. We also introduce new Gaussians by enabling the densification procedure detailed in [8]. Finally, each Gaussian is parented to the closest particle and its location relative to the particle is stored as a bond. Any Gaussian that is farther from a threshold to a particle is discarded. This reconstruction process as well as its results are visualized in Fig. 2. A typical scene contains around a thousand particles and ten thousand Gaussians.

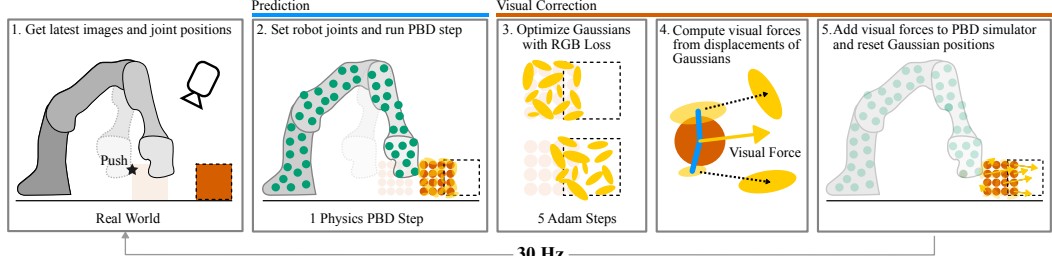

Figure 3: Our real-time correction method illustrated in its different steps.

The robot is modelled similarly using renders from its known meshes and inserted into each scene. The background elements (a white table in our experiments) are modelled only using Gaussians with the conventional training regime [8].

## 3.2 Online Prediction and Correction

After the initialization phase, our approach employs a combination of Position-Based Dynamics (PBD) and Gaussian splatting optimization to predict the current state of our representation. Our method can be decomposed into two stages: a prediction stage and a correction stage. These two stages are called sequentially and are illustrated in Fig. 3.

**Prediction** The PBD physics system acts upon the particles in our system at a rate of 30Hz. The prediction step constitutes setting the particles associated with the robot to the positions calculated by the forward kinematics, running a single PBD physics step that forward projects the particles and resolves physical constraints (as described in Sec. 2), and finally rigidly moving the Gaussians bonded to particles to their new predicted locations. In our system, this takes approximately 5 ms to perform on an NVIDIA 3090, leaving 28 ms for the subsequent correction stage.

**Correction** After the physics prediction, the current state is rendered from known camera viewpoints. The renders are compared to the images received from the cameras and the photometric reconstruction loss is reduced by optimizing the parameters of the Gaussians. Note that only RGB data is required for the optimization. All Gaussians are allowed to learn new colors and opacities at a low learning rate. This allows shadows imposed by the robot and by the objects to be explained by changing the colors of the Gaussians. Only Gaussians attached to objects are allowed to change their positions and orientations. After 5 optimization steps – a number tuned to meet realtime constraints – the desired positions of the Gaussians are stored and all Gaussians are **reset** to their original position before the optimization started. The difference in the desired positions of the Gaussians and the original positions of the Gaussians is used to compute a force which is imposed on the connected particles. This force is calculated as $\boldsymbol{f}_i = K_p \sum_j o_j (\boldsymbol{g}_j - \boldsymbol{g}_j^0)$, where $\boldsymbol{g}_j$ and $\boldsymbol{g}_j^0$ are the final and initial positions of the Gaussians connected to particle $i$, $o_j$ is the opacity, and $K_p$ is a proportional gain that has to be tuned (We found $K_p = 60$ to be a good value for our experiments). The Gaussians are thus never directly moved by the correction step. Rather, the correction step is used to generate corrective external forces on the particles which are ultimately resolved by the physics system. The resulting movement of the particles, as orchestrated by the physics, causes the Gaussians bonded to them to move. Consequently, the system is never in a physically infeasible state. Moving the Gaussians in this manner ensures that they always remain in place relative to their bonded particles.

## 4 EXPERIMENTS

We rigorously evaluate the performance of our proposed system across several key metrics to determine its efficacy in dynamic object tracking and photometric reconstruction from novel viewpoints.

**Datasets** To evaluate our method, we utilize both a simulated dataset and a real dataset, each comprising a tabletop scene. The simulated dataset (Fig. 4) consists of 25 scenarios designed to highlight various dynamic conditions, including single object pushing (5 scenes), multiple object pushing (5 scenes), object pickup (5), object pushover (5), and pushing a deformable rope (5). The real dataset (Fig. 5) contains 25 scenarios exhibiting similar variations in dynamic conditions. The real-world experiments have Aruco markers attached to the objects. The markers are used to extract

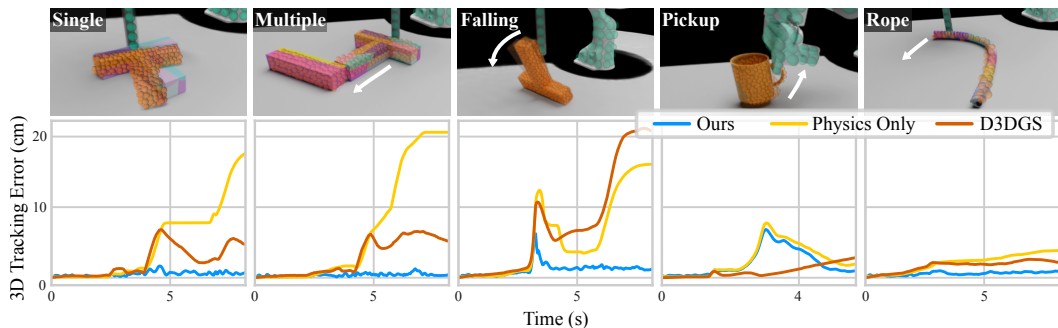

Figure 4: Tracking error on a set of points attached to moving objects on synthetic scenes showing different dynamic conditions that include pushing, falling, picking up, and deformable objects.

Table 1: Tracking error and photometric reconstruction quality from unseen views on the **simulated dataset** for our full method, physics only, augmented D3DGS and Cotracker [19]

| | ↓ 3D Tracking error (cm) | | | ↓ 2D Tracking error (px) | | | | ↑ PSNR | | |
|---|---|---|---|---|---|---|---|---|---|---|
| | Ours | Physics | D3DGS* | Ours | Physics | D3DGS* | [19] | Ours | Physics | D3DGS* |
| Single | **0.65** | 9.87 | 4.05 | **5.46** | 81.44 | 36.19 | 29.13 | **17.32** | 14.16 | 16.52 |
| Multiple | **0.52** | 6.70 | 2.71 | **5.92** | 52.84 | 14.25 | 17.40 | **17.52** | 14.34 | 17.07 |
| Falling | **1.52** | 11.00 | 7.64 | **14.50** | 117.10 | 62.93 | 48.33 | **16.79** | 12.88 | 16.26 |
| Pickup | 2.25 | 2.58 | **0.93** | 27.61 | 31.49 | **9.95** | 14.24 | 13.88 | 13.17 | **15.84** |
| Rope | **1.02** | 4.50 | 2.71 | **7.08** | 25.55 | 22.50 | 28.75 | 14.63 | 11.97 | **15.47** |

groundtruth 2D and 3D trajectories using a combination of Aruco detection, manual labelling, and factor-graph based optimization [18]. For both datasets, we employ 3 cameras to derive the visual forces, while 2 cameras are used for evaluation purposes only. This diverse set of experiments allows us to assess the performance of our method under a range of challenging scenarios, encompassing both rigid and deformable objects, as well as interactions involving single and multiple objects.

**Baselines** A baseline that is realtime, correctable and serves as world model could not be found – therefore, we compare our method with baselines that can **separately** track and predict. We note that our main contribution is that we can do both at the same time. Therefore, we compare our method as a 3D tracker against Dynamic 3D Gaussians (D3DGS) [9], a 2D tracker against Cotracker [19], and as a forward model against only using a physics simulator [5] without visual forces.

Unlike our approach, D3DGS incorporates physical priors directly into the Gaussian optimization process through auxiliary losses. D3DGS cannot be used as world model like our method since it cannot project what may happen to the Gaussians if forces act on the system. It also requires a foreground mask to be provided at each timestep. In [9], background subtraction is used to acquire those masks. However, we found that under our challenging realtime constraints where only 3 cameras are used – this leads to catastrophic failure in tracking as visualized in Fig. 6. Therefore, to make it competitive, we augment the baseline (D3DGS*) by giving it groundtruth hand-labelled masks and by forcing the Gaussians associated with the robot to move according to the forward kinematics of the robot rather than through the optimization process. Furthermore, we found D3DGS auxilliary losses make it half as slow as our visual training iteration, but we assume that optimizations can be made to match the speed of ours and thus allow them six training iterations to match our five training iterations and one physics step. For our method, the full parameters are listed in the supplementary.

**Metrics** We evaluate our method on the mean error in the 2D and 3D trajectories of known points. We also evaluate the foreground photometric reconstruction quality (which includes only the objects on the table) from unseen viewpoints. The predicted 3D trajectory of a query point is obtained by tracking the frame of the Gaussian that was closest to that query at the first timestep. This procedure is consistent with the approach used in [9]. 3D trajectories are projected into each camera to obtain the 2D trajectories and their initial points are used to query Cotracker [19]. The trajectory error is calculated as the mean difference between the groundtruth and the predicted trajectories of several points sampled on the objects.

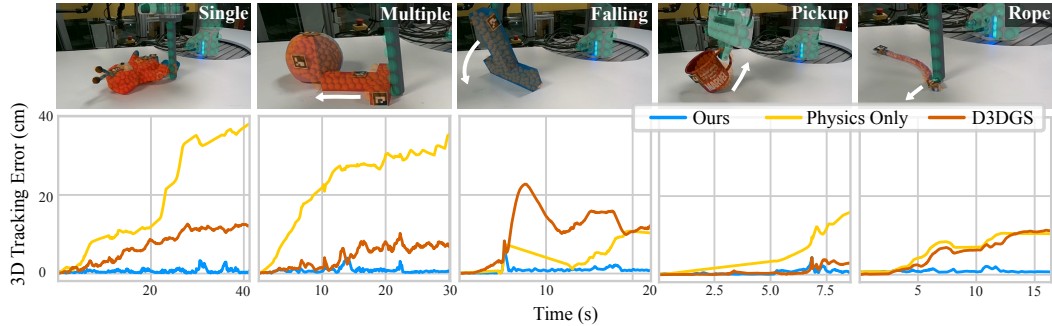

Figure 5: The tracking error on a set of points attached to moving objects is recorded on real scenes.

Table 2: Tracking error and photometric reconstruction quality from unseen views on the **real dataset** for our full method, physics only, augmented D3DGS and Cotracker [19].

|  | ↓ 3D Tracking error (cm) | | | ↓ 2D Tracking error (px) | | | | ↑ PSNR | | |
|  | Ours | Physics | D3DGS* | Ours | Physics | D3DGS* | [19] | Ours | Physics | D3DGS* |
|---|---|---|---|---|---|---|---|---|---|---|
| Single | **1.40** | 31.71 | 5.96 | **11.90** | 264.20 | 57.10 | 88.42 | **16.49** | 10.46 | 12.05 |
| Multiple | **1.84** | 26.07 | 10.09 | **17.00** | 238.42 | 57.75 | 110.79 | **16.99** | 10.64 | 14.18 |
| Falling | **2.06** | 7.80 | 5.90 | **16.51** | 62.56 | 56.98 | 85.33 | **17.05** | 13.56 | 15.87 |
| Pickup | **0.68** | 3.36 | 3.36 | **6.65** | 37.34 | 7.80 | 8.21 | **16.38** | 13.40 | 15.22 |
| Rope | **1.01** | 11.24 | 11.24 | **6.68** | 76.98 | 82.45 | 41.87 | **15.81** | 13.73 | 12.11 |

**Results** Fig. 4 and 5 show 5 of the 25 scenarios tested for each of the simulated and real datasets. The 3D tracking error is plotted over time and shows our method robustly tracking the objects in the scene. Tables 1 and 2 summarize our metrics over each scenario. Our method outperforms all baselines on all experiments except the simulated Pickup tasks. The Pickup tasks are highly dynamic environments where our physical priors were significantly different to the physics exhibited in the simulated scene (see videos on website). This highlights an expected weakness of our system where significantly misaligned physical priors can degrade performance. Nevertheless, our system is able to recover and acquire the final state of the pickup task as shown in Fig. 4. In all other experiments (45/50), physical priors significantly improve tracking performance. Fig. 6 shows qualitative results.

## 5 RELATED WORK

To the best of our knowledge, we are the first to create a representation consisting of both particles and 3D Gaussians for the purposes of fusing physical and visual priors within a correctable robotics world model. However, the use of a particle-based physics system alongside a visual component other than Gaussians has been at the core of other works [20, 21, 22, 23]. Moreover, regularizing Gaussian splatting with physical priors through means other than a physics framework has also been explored [9] (the baseline used in our experiments).

ParticleNeRF [20] uses particles that are acted upon by a physics system and can be rendered from any viewpoint using a Neural Radiance Field [24] formulation. Particles are associated with latent features which can be decoded into a radiance field by a small neural network. While ParticleNeRF introduced the use of a particle-based physics system (PBD) to incorporate physical constraints and deviations from the NeRF's reconstruction loss relative to particle positions, it only utilized collision constraints and did not fully explore the idea of employing various physical priors to regularize the reconstruction. In part, due to the slower NeRF formulation and the requirement of upwards of 10 cameras to constrain the optimization, ParticleNeRF could not be deployed in a real world robotics setting with realtime constraints. In contrast, our work uses the much faster Gaussian splatting to represent the visual world and further makes heavy use of physical priors which allow visual corrections from as little as 3 cameras in the scene.

Particles within a PBD framework paired with corrections from observed pointclouds have been used to model soft human tissue within the domain of surgical robotics [21, 22, 23]. Our method is more general, realtime, and uses visual feedback achieved through the fast Gaussian splatting rather than pointclouds and SDFs.

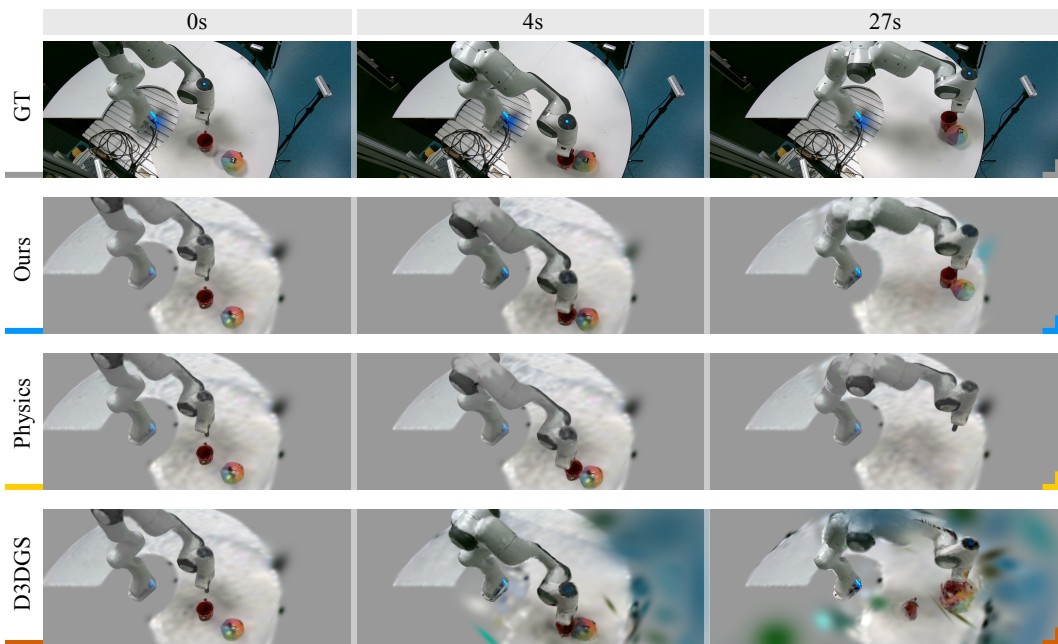

Figure 6: A scene's predicted visual state with different methods. Our method synchronizes with the real state using a combination of physical prediction and visual correction. Physical prediction alone can estimate the state for about 2 seconds but will eventually desynchronize. Visual correction alone (D3DGS) allows the Gaussians to move in physically infeasible ways, creating scenarios where objects split and Gaussians move freely within the object despite auxiliary structural losses.

D3DGS [9] adds physical priors to the Gaussian optimization process with the aim of regularizing the movement of the Gaussians. The significant difference with our work is that the physical priors are not strictly enforced by a physics system – rather auxiliary losses are added to the photometric reconstruction loss. Importantly, this means that D3DGS cannot act as a world model because it cannot be used to predict future states. Nevertheless, D3DGS has shown good tracking and reconstruction performance when groundtruth masks are provided or when upwards of 20 well-distributed cameras are observing the scene and the optimization is given ample time to converge. While these requirements are difficult to achieve in a robotics setting, this visually-driven optimization serves as our baseline and ablates the visual component of our system as well as outlines the importance of grounding the optimization with a physics system.

## 6   LIMITATIONS AND CONCLUSIONS

Our method was demonstrated on a table-top scenario where heavy use of physical priors can be made. The extension to an open-world setting is left to future work. We make an assumption that the predicted state will be close enough to the groundtruth so that visual forces can be effective. This assumption is broken when significant errors in modelling cause the physics system to accrue errors at a faster rate than can be corrected. This can occur when an object is moved very quickly by the robot. Consequently, a method of closing the loop using global visual information is needed – plausibly using a system like [25]. We foresee that a more sophisticated initialization procedure that uses learnt shape priors would be more powerful. Lastly, our method does not alter the physical structure or parameters of the objects once the modelling is done. Therefore, new observations are not used to correct modelling mistakes. A means of online structure correction and system identification would extend this framework.

In conclusion, we have presented a hybrid representation consisting of particles and 3D Gaussians that together represent the physical and visual state of the world. In conjunction, they can be used to predict future states and correct the predicted state from observed data. This synergy makes them suitable for use as a world model in future robotic works. The world model can then be used to extract object state for a reinforcement-based or an imitation-based policy or to plan for future actions using model predictive control.

## ACKNOWLEDGMENT

The authors acknowledge continued support from the Queensland University of Technology (QUT) through the Centre for Robotics. This work was partially supported by the Australian Government through the Australian Research Council's Discovery Projects funding scheme (Project DP220102398).

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

# Supplementary Material
# Physically Embodied Gaussian Splatting: A Visually Learnt and Physically Grounded 3D Representation for Robotics

Jad Abou-Chakra[1], Krishan Rana[2], Feras Dayoub[1], Niko Sünderhauf[1]

[1]Queensland University of Technology
[2]University of Adelaide

## A   Experimental Setup

The real-world experiments are conducted using the tabletop setup shown in Supp. Figure 1. The setup employs a Franka Emika robot equipped with two end-effectors: a standard gripper for pick-up scenarios and a pusher for other scenarios. The tabletop and robot are observed by five cameras: three Intel RealSense D455 cameras and two D435 cameras. These cameras are jointly calibrated using a hand-eye calibration technique. During operation, all five cameras are utilized for system initialization. However, only the three D455 cameras are employed during the prediction and correction stages. In all scenarios, the robot is teleoperated to manipulate objects on the tabletop. The datasets are captured by recording the image stream from the cameras and encoding them as HEVC videos. These videos are scaled to a resolution of 640x360 and decoded in real-time during evaluation to mimic live operation. Additionally, the robot's joint positions are timestamped and saved during the recording process and replayed during the evaluation.

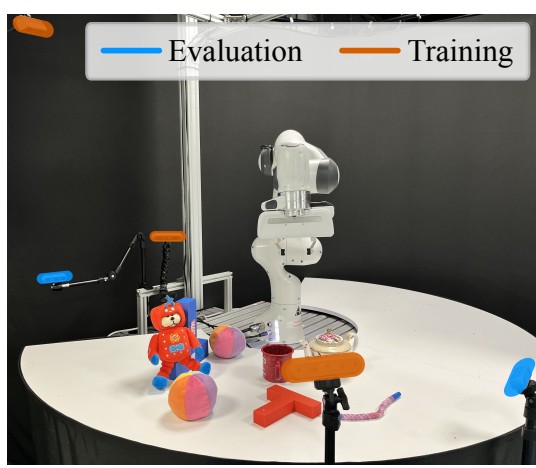

Supplementary Figure 1: The tabletop setup used in the real experiments showing the robot, some of the objects used in the scenarios, and the position of the 5 cameras used.

## B   Implementation

The system follows a two-step process: initialization and prediction/correction. During initialization, particles and Gaussians are generated for each detected object in the scene. Subsequently, the system enters the prediction and correction stage, where the particles are simulated using a Position-Based Dynamics (PBD) physics system, while corrective forces are calculated based on the Gaussians attached to the particles. This section elaborates on the implementation and parameterization details of each phase.

**Static Scene Initialization**   The tabletop is modeled using the five RGBD cameras in the scene, employing the standard Gaussian Splatting technique. However, to avoid interference with object placed on the table, the Gaussians are initialized as thin disks. Additionally, the table's pointcloud is utilized to calculate the ground plane. The Gaussians are trained using the Adam optimizer for 500 steps, with a position learning rate of $1e^{-4}$, color learning rate of $2.5e^{-3}$, scaling learning rate of $1e^{-3}$, opacity learning rate of $1e^{-2}$, and rotation learning rate of $1e^{-3}$. The scale is clamped between 1 mm and 1 cm.

**Robot Initialization** The robot's particles are manually fitted to the links using Blender. The link each particle belongs to is stored so that forward kinematics can be used to appropriately change its position. Furthermore, the robot is rendered in Blender from multiple viewpoints, and Gaussians are trained to reconstruct these renders. These Gaussians are then bonded to the closest particle on the robot. The combination of particles, Gaussians, and bonds is inserted at the start of every scenario. The same parameters used for training the static scene are also applied to the robot.

**Object Initialization** For each object, the 3D bounding box is calculated from its pointcloud, which is extracted from the depth and instance masks. The initialization process is described in Algorithm 1. We use $n = 80$ and $m = 250$. All particles are initialized to a mass of 0.1 kg with the exception of the real and simulated rope which are set to 0.2 kg and 0.3 kg. The higher the mass of the particle, the less the influence of the visual force. The mass acts as both a *physical* and a *visual* inertia. These concepts can be separated in future work if fine-grained tuning is needed. For scenarios involving rope, the corrective forces are less reliable than that of larger bodies because they occupy less pixels in the image and the physical priors are less constraining because of the deformability. We compensate for the increased noise in the corrective forces by increasing the visual inertia. Note that Algorithm 1 is repeated for each object. Future work may choose to build all the objects simultaneously rather than sequentially to reduce the overall duration of the initialization. In the current implementation, object modeling takes approximately 20 to 40 seconds, which we found acceptable given that it is only done once per scenario.

**Prediction Step** The Position-Based Dynamics (PBD) physics system is used to predict the locations of the particles and the Gaussians at each timestep. It runs at a fixed frequency of 30 Hz (33.33 ms per step). The physics step is described in Algorithm 2. We use 20 substeps. At each substep, the velocities and forces are integrated, and then the constraints are solved using a Jacobi solver. Four Jacobi iterations are employed to sufficiently solve the physical constraints. After every physics step, the particle velocities are multiplied by 0.9 (an empirically chosen value). This damping contributes to system stability.

---

**Algorithm 1** Dual Gaussian-Particle Initialization

1: Fill BBox with Spherical Gaussians
2: Prune Gaussians Not In Instance Masks
3: **for** n iterations **do**
4:      **for** all images and masks $I$ **do** ▷ Adam step
5:          $L \leftarrow L_{\text{rgb}} + L_{\text{seg}}$
6:          L.backward()
7:          $\boldsymbol{g}, a, \boldsymbol{c}$ = optimizer.step()
8:      **for** k iterations **do**          ▷ Jacobi step
9:          $\boldsymbol{g}$ = solveCollisionConstraints($\boldsymbol{g}$)
10:          $\boldsymbol{g}$ = solveGroundConstraints($\boldsymbol{g}$)
11: $\boldsymbol{p} = \boldsymbol{g}$ ▷ Create particles at Gaussian locations
12: Initialize particle mass and velocities
13: Create particle shape constraints
14: **for** m iterations **do**
15:      **for** all images and masks $I$ **do**
16:          $L_{\text{rgb}}$.backward()
17:          $\boldsymbol{g}, a, \boldsymbol{c}, \boldsymbol{s}$ = optimizer.step()
18:          $\boldsymbol{g}, a, \boldsymbol{c}, \boldsymbol{s}$ = densify($\boldsymbol{g}, a, \boldsymbol{c}, a$)
19: **for** each Gaussian i **do**
20:      $g_i$.parent = findClosestParticle($\boldsymbol{p}$)

---

**Algorithm 2** PBD Physics Step

1: **for** all particles $i$ **do**          ▷ Integrate particles
2:      $\boldsymbol{p}_0, \boldsymbol{q}_0 \leftarrow \boldsymbol{p}_i, \boldsymbol{q}_i$
3:      $\boldsymbol{p}_i \leftarrow \boldsymbol{p}_i + \Delta t \boldsymbol{v}_i + \frac{\Delta t^2}{m_i}(\boldsymbol{f}_i + \text{gravity})$
4:      $\theta \leftarrow \frac{|w_i| \Delta t}{2}$
5:      $\boldsymbol{q}_i \leftarrow [\frac{\tilde{\boldsymbol{\omega}}_i}{|\boldsymbol{\omega}_i|} \sin \theta, \cos \theta] \boldsymbol{q}_i$
6: **for** k solver iterations **do** ▷ Resolve constraints
7:      **for** all particles i **do**
8:          $\boldsymbol{p}_i \leftarrow$ groundConstraints($i$)
9:      **for** all collision pairs i, j **do**
10:          $\boldsymbol{p}_i \leftarrow$ collisionConstraints($i, j$)
11:      **for** all shapes s **do**
12:          **for** particles i in s **do**
13:              $\boldsymbol{p}_i, \boldsymbol{q}_i \leftarrow$ shapeMatching($i, s$)
14: **for** all particles $i$ **do**          ▷ Update velocities
15:      $\boldsymbol{v}_i \leftarrow (\boldsymbol{p}_i - \boldsymbol{p}_0)/\Delta t$
16:      $\boldsymbol{\omega}_i \leftarrow \text{axis}(\boldsymbol{q}_i \boldsymbol{q}_0^{-1}).\text{angle}(\boldsymbol{q}_i \boldsymbol{q}_0^{-1})/\Delta t$

---

**Algorithm 3** Visual Forces

1: $\boldsymbol{g}^{\text{prev}} \leftarrow \boldsymbol{g}$          ▷ Save positions
2: o = AdamOptimizer()
3: **for** n iterations **do**
4:      Choose random image $I$
5:      $L_{\text{rgb}}(I)$.backward()
6:      $\boldsymbol{g}$[not objects].grad = 0
7:      $\boldsymbol{g}, \boldsymbol{c}, \boldsymbol{o}, \boldsymbol{R} \leftarrow$ o.step()
8: **for** every Gaussian $i$ **do**
9:      k = $g_i$.parent
10:      **if** k is not None **then**
11:          $\boldsymbol{f}_k \leftarrow \boldsymbol{f}_k + K_p(\boldsymbol{g}_i - \boldsymbol{g}_i^{\text{prev}})$
12: $\boldsymbol{g} \leftarrow \boldsymbol{g}^{\text{prev}}$          ▷ Reset positions

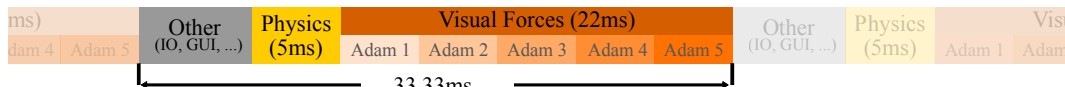

Supplementary Figure 2: The various functions called during the prediction and the correction step profiled. In the 'Other' phase, the GUI is drawn and new sensor observations are read. The physics step takes 5 ms and is followed by approximately 22 ms of Adam optimizations that are used to compute the visual forces.

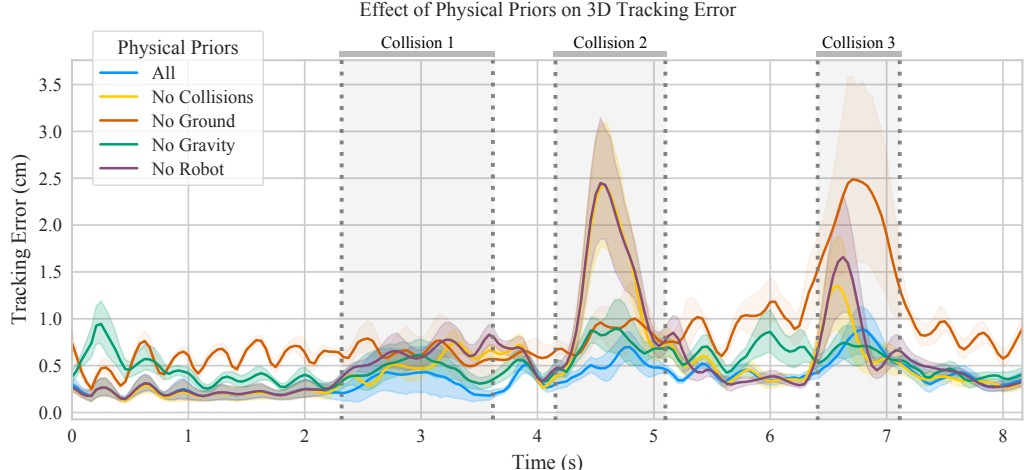

Supplementary Figure 3: An ablation showing the effect of different physical priors on the 3D tracking error of 12 points located on two objects on a tabletop. The scene used for this ablation is "Multiple1" from the simulated dataset. Using all physical priors produces on average the lowest tracking error over time.

**Correction Step**  Visual forces are computed in the correction step using Algorithm 1. Gaussian displacements are calculated using 5 iterations of the Adam optimizer. The scales of the Gaussians are fixed, while the positions, rotations, opacities, and colors are allowed to change. Adam's internal parameters are reset at every new physics step. Gaussian displacements below 2 mm are ignored to increase stability. The position learning rate is set between $1e^{-3}$ and $3e^{-3}$ depending on the scene, while the rotation, color, and opacity learning rates are set to $1e^{-4}$, $5e^{-4}$, and $5e^{-4}$, respectively. Allowing colors, opacities, and rotations to change gives the system more ways to explain lighting variations that should not be explained by the motion of the Gaussians. A $K_p$ of 60 is used in all scenarios.

The prediction and correction steps are profiled in Supp. Figure 2.

## C  Ablations

**Physical Priors**  We evaluate the effectiveness of our system's embedded physical priors by simulating a scene with two objects, as illustrated in Supp. Figure 3 and summarized Supp. Figure 5. The scenarios highlight how our system's performance is enhanced by incorporating various physical constraints: (i) With all physical priors enabled, our system accurately captures the objects' dynamics, including collisions and interactions with the environment. (ii) When collisions between particles are ignored, the objects' states deviate from the ground truth, particularly during intense collision events (Collisions 2 and 3). (iii) Disabling the ground plane and gravity causes the objects' motions to oscillate continuously, as their movements are no longer properly regulated. (iv) Even with the ground plane intact, disabling gravity leads to similar oscillatory behavior, as the objects are not subjected to the expected downward force. By adding physical priors, our system achieves better predictions that more closely match the groundtruth.

Supp. Figure 4 ablates (i) the number of cameras used to compute visual forces, (ii) the resolution of the images used for the reconstruction loss, (iii) the effect of visual gain, (iv) the Gaussian position learning rate, and (v) the number of Adam iterations. The mean tracking error across the entire trajectory is also reported in Supp. Figure 5.

**Cameras** The ablations reveal that increasing the number of cameras yields diminishing returns within our framework. We observe that higher resolutions lead to lower tracking error. There is, however, only a slight difference between 1280x720, 640x360, 320x180. 1280x720 comes at a significant computational cost with visual force computation taking approximately 40 ms compared to the 20 ms for the lower resolutions. Below 640x360, the factor limiting performance is no longer resolution and thus there is no performance gain. For these reasons, we choose the 640x360 as the image size with which we calculate the visual forces.

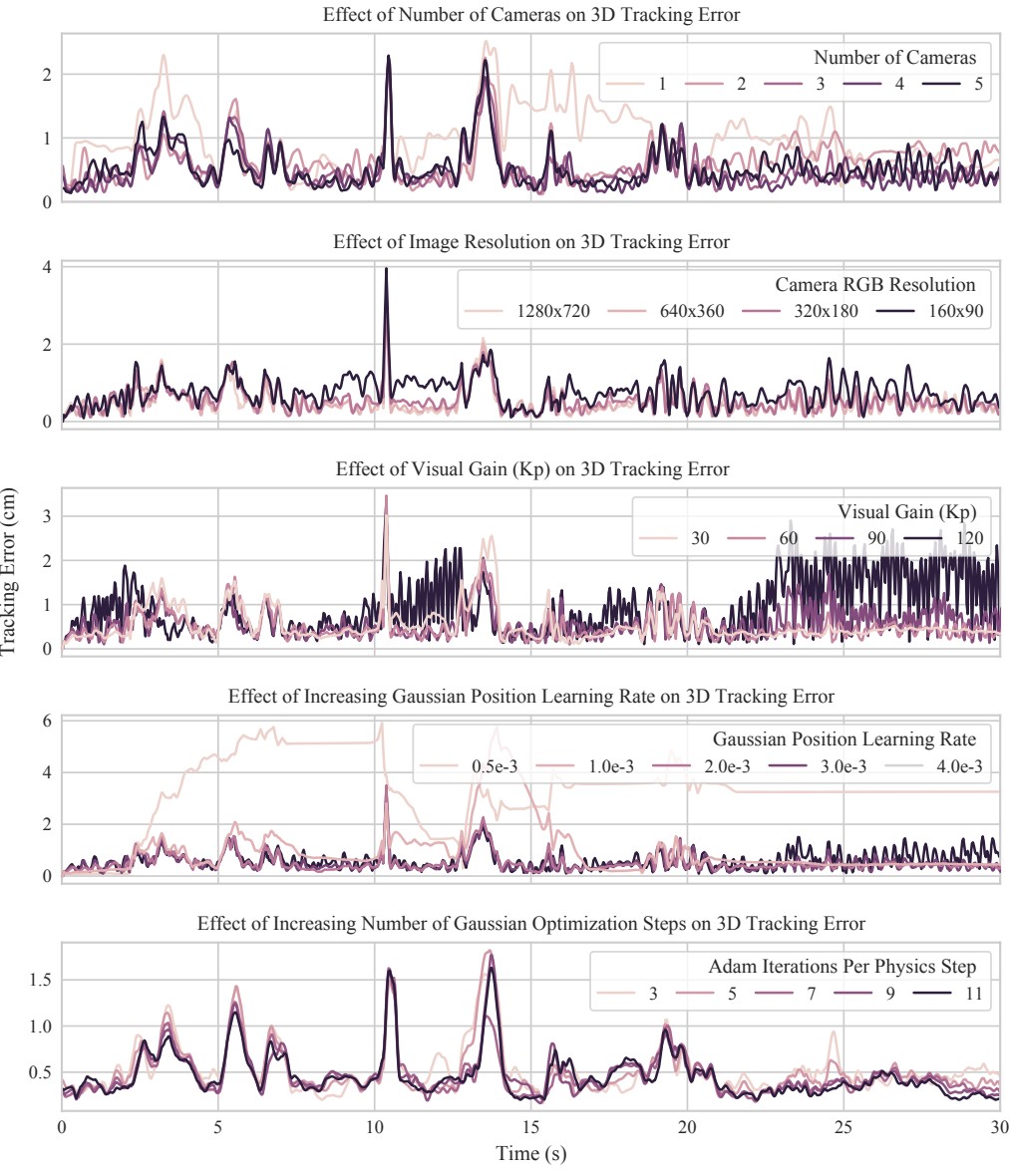

Supplementary Figure 4: The effect of varying the parameters of our system on 3D tracking performance.

| Physical Priors | All | No Collision | No Gravity | No Robot | No Ground |
|---|---|---|---|---|---|
| 3D Tracking error (cm) | 0.36±0.27 | 0.52±0.58 | 0.52±0.33 | 0.57±0.61 | 0.84±0.72 |

(a) Effect of Physical Priors on the 3D Tracking Error in cm reported as a mean±standard deviation.

| Number of Cameras | 1 | 2 | 3 | 4 | 5 |
|---|---|---|---|---|---|
| 3D Tracking Error (cm) | 1.16 ±0.82 | 0.41±0.47 | 0.37±0.27 | 0.4±0.28 | 0.39±0.28 |

(b) Effect of Number of Cameras on the 3D Tracking Error in cm reported as a mean±standard deviation.

| Image Resolution | 1280x720 | 640x360 | 320x180 | 160x90 |
|---|---|---|---|---|
| 3D Tracking error (cm) | 0.58±0.49 | 0.59±0.49 | 0.65±0.50 | 0.87±0.56 |

(c) Effect of Number of Cameras on the 3D Tracking Error in cm reported as a mean±standard deviation.

| Visual Forces Gain | 30 | 60 | 90 | 120 |
|---|---|---|---|---|
| 3D Tracking error (cm) | 0.41±0.41 | 0.40±0.33 | 0.460±0.37 | 0.750±0.57 |

(d) Effect of Visual Gain (Kp) on the 3D Tracking Error in cm reported as a mean±standard deviation.

| Learning Rate | 0.5e-3 | 1e-3 | 2e-3 | 3e-3 | 4e-3 |
|---|---|---|---|---|---|
| 3D Tracking error (cm) | 7.10±8.53 | 0.55±0.48 | 0.40±0.36 | 0.39±0.31 | 0.40±0.31 |

(e) Effect of Increasing Gaussian Position Learning Rate on the 3D Tracking Error in cm reported as a mean±standard deviation.

| Number of Adam Iterations | 3 | 5 | 7 | 9 | 11 |
|---|---|---|---|---|---|
| 3D Tracking error (cm) | 7.1±8.53 | 0.62±0.59 | 0.40±0.36 | 0.40±0.38 | 0.43±0.46 |

(f) Effect of Increasing Number of Gaussian Optimization Steps on the 3D Tracking Error in cm reported as a mean±standard deviation.

Supplementary Figure 5: The effect of varying the parameters of our system on the **mean** 3D tracking performance.

**Visual Forces**   Our framework uses visual forces to create the corrective actions needed to keep the Gaussian-Particle representation synchronized. This results in smooth corrections but it also creates dynamic effects that, without careful tuning, creates oscillations. These oscillations are similar to the behaviour of an undamped spring system. Future work may look into removing the oscillatory effect by possibly adding a derivative term to the visual force calculation. For this work, we tune our system and find a balance between an acceptable amount of oscillations and tracking ability. The ablations in Supp. Figure 4 show that a high gain (and/or a high Gaussian position learning rate) produces high oscillation and that a low gain (and/or a low learning rate) has a detrimental effect on tracking. The number of Adam iterations is chosen so that realtime constraints are met. The ablation shows that reducing the number of Adam iterations is a trade off that can be made when the physics timestep takes longer than expected without a significant impact on the overall synchronization of the world model.

**Initialization**   We evaluate our initialization method by comparing it with particle positions manually modeled using Blender. We refer to the manual initialization as the "oracle" initialization because the known object meshes are used to manually model the objects with particles. We show examples of the oracle initialization alongside our method's initialization in Supp. Figure 6. We evaluate the tracking performance of our system using the oracle initialization and the initialization scheme described in our method. The results, presented in Table 1, demonstrate that our automated initialization method achieves performance comparable to the oracle.

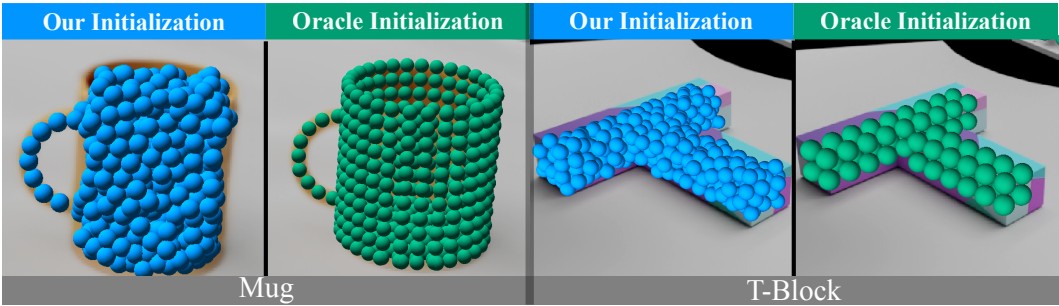

Supplementary Figure 6: Particle placements of our initialization method compared to the oracle initialization.

Supplementary Table 1: Tracking error and photometric reconstruction quality from unseen views from a scene from each scene category. The table compares our automatic initialization with a manual modelling procedure (oracle), showing similar tracking errors.

| | 2D Tracking Error (px) | | 3D Tracking Error (cm) | | PSNR (FG) | |
|---|---|---|---|---|---|---|
| | Ours | Oracle | Ours | Oracle | Ours | Oracle |
| Single | 5.31 | 4.27 | 0.54 | 0.44 | 17.69 | 17.74 |
| Multiple | 4.38 | 6.69 | 0.39 | 0.55 | 17.73 | 17.65 |
| Falling | 13.04 | 14.61 | 1.18 | 1.52 | 16.18 | 16.65 |
| Pickup | 25.92 | 28.89 | 2.09 | 2.25 | 15.11 | 14.65 |
| Rope | 10.96 | 9.62 | 1.39 | 1.07 | 14.73 | 15.55 |

## D   Failure Modes

The Gaussian-Particle representation can deviate from the groundtruth in several ways. If the rendered state of the scene significantly differs from the groundtruth image, the visual forces will not create meaningful corrections.

Additionally, if the physical modelling is significantly different to its real world counterpart, the physical priors will have a detrimental effect on the tracking performance of the system. This can be seen in the real scenario title "Pushover 5" in Supp. Figure 7 where a T-Block could not be pushed over and thus escaped the radius of convergence of the visual forces.

In some instances, both the texture and the geometry of the object are simultaneously ambiguous. In the simulated "Rope 1" scenario, the rope can rotate around its spine without impacting either the geometry or the texture thus allowing for a slight steady state error to occur.

## E   Design Choices

The primary utility of our representation is that it can be forward simulated at faster than realtime rates and corrected from visual observations at realtime rates. Our design choices were thus primarily guided by the need to lower the computational overhead of both the prediction and the correction step.

We opted for Position-Based Dynamics (PBD) despite its lower physical accuracy compared to other simulators because of its superior real-time performance and stability. Additionally, we chose Gaussian Splatting over other differentiable rendering techniques due to its status as the fastest differentiable renderer available.

The decision to use visual forces for state correction was also driven by the need to reduce computational demands. Although alternative methods, such as integrating additional shape matching constraints between the particles and the Gaussians, could allow Gaussians to influence particle positions, these approaches would be more computationally expensive than our method. They would require resolving constraints over multiple iterations in every PBD substep and would couple the physics system with the correction system, making performance of the physics loop dependent on

1. Physical Prediction Failure | 2. Visual Symmetries

Pushover 5 | Rope 1

GT Trajectory — Tracked Trajectory

Supplementary Figure 7: The first image shows a highly dynamic scenario where the physics failed to push the TBlock into a location where visual forces could correct it. The second image shows a scenario where both visual and geometrical symmetries allowed the rope to rotate around its central axis and created a steady state error in tracking.

the number of Gaussians in the scene. In contrast, our method uses visual forces that are simple to implement, fast, and maintain a clear separation between prediction and correction systems.

## F  Experimental Results

The 3D tracking performance of our system on all scenarios are shown in Supp. Figure 8.

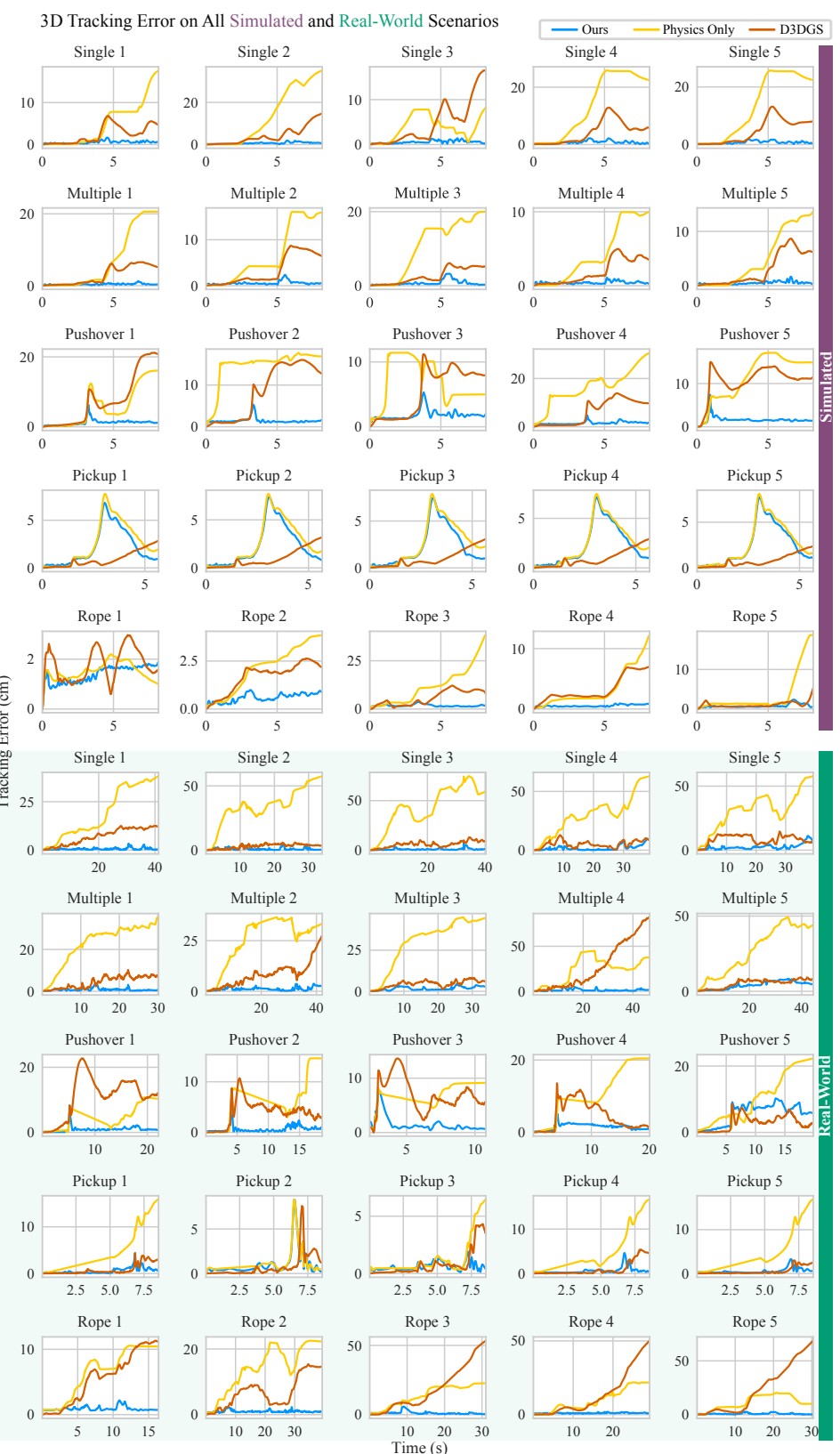

Supplementary Figure 8: The 3D tracking performance of our system and its baselines on all scenarios (simulated and real)

