# OpenReview forum: "Physically Embodied Gaussian Splatting: A Visually Learnt and Physically Grounded 3D Representation for Robotics"
_robot-learning.org/CoRL/2024/Conference — CoRL 2024_

### Official Review · Reviewer_iPN3 · 2024-07-18
**Interesting work, needs more ablations**

**Originality:** 4
**Technical Quality:** 4
**Clarity Of Presentation:** 5
**Potential Impact:** 4
**Recommendation:** 4
**Confidence:** 3

**Review:**

Strengths:
- The idea of combining the strengths of Gaussian Splatting and PBD is an interesting combination that allows for predictive capabilities and visual corrections. The proposed method demonstrates this quite well, and quantitative experiments show that these physical predictions can help tracking in scenarios where robots are interacting with rigid/deformable objects. As mentioned in the paper, no baseline exists that is both physically predictive and capable of being corrected via visual feedback, thus the representation's novelty is significant.
- The real-time demo system for tracking is quite impressive. It has been well-engineered. Additionally, the conducted experiments demonstrate better tracking in both 2D and 3D compared to a baseline without visual feedback (which demonstrates the utility of the visual feedback component to the system), and against the D3DGS method.
- The paper is well-written and easy to follow.


Weaknesses:
- While the current ablation demonstrates the utility of the visual feedback component of the system, there are many more components in the system that are not ablated. The main weakness is that the paper lacks ablations to understand many of these design choices. For example, the initialisation step has many details (such as optimising Gaussians as particles then separating them) that are not well-understood, and it's not clear how important this initialisation step is for performance. Readers cannot decouple these many design choices to the overall performance of the tracking. Other examples include: why is the visual feedback distilled into a visual force? Why not just quickly optimise the particle positions based on the Gaussian deformations?
- Minor weakness: as the title of the paper mentions the representation is a world model, it would have been nice to see downstream applications of this representations such as reinforcement learning.

**Quality Of The Limitations Section:**

3

**Questions For Rebuttal:**

Questions/Comments:
- What are the details of the shape matching constraints?
- For the prediction step (PBD), I assume there is a given joint torque applied to the robot by a user? Can the authors detail this?
- Line 147: What are "features" here? did authors mean the positions/orientations/scale/color?
- While the supplement has some ablations that show results on a single video, quantitative results could be helpful here.

**Robotics Focus:**

4

**Summary Of Paper:**

This paper proposes a representation based on particle-based dynamics (PBD) and Gaussian splatting that is both predictive and can be corrected by visual observations in real-time. A detailed initialisation scheme is proposed to initialise both the particles and Gaussians along with the rigid "bonds" that connect them. Prediction is done via particle-based dynamics, and corrections use a few Gaussian splatting optimisation steps to computed a "visual force" to be used within the PBD to correct the particle positions. Experiments demonstrate better tracking than baselines, and demos/videos demonstrate impressive tracking of rigid and linearly deformable objects (e.g. ropes).

**Summary Of Recommendation:**

POST-REBUTTAL UPDATE: Really interesting. The work has been scoped more properly, and more ablations in the supplement provide insight into the strength of the initialisation. I do wish there was more insight into the design choices of the system, but I think the contributions are important enough to be shown at CoRL.

---

### Official Review · Reviewer_ud25 · 2024-07-20
**Questionable contribution**

**Originality:** 2
**Technical Quality:** 3
**Clarity Of Presentation:** 4
**Potential Impact:** 2
**Recommendation:** 3
**Confidence:** 4

**Review:**

This paper proposed a novel approach for achieving the robotic world model. A particle-based physics system to anticipate the future plausible states and a Gaussian Splatting module to leverage visual observations to online correct the predictions. The proposed method is evaluated on 2D and 3D tracking tasks and photometric reconstruction quality.

### Strengths:
* This paper proposed a `Gaussian-Particle` bonds to couple the two representations to enable the visual-driven physical prediction correction, which is interesting and novel to the best of my knowledge.
* The proposed approach leverages the particle-based physics simulation to introduce several constrains to ensure the prediction is physically feasible.
* The proposed method is evaluated on 3D tracking, 2D tracking, and physical simulation tasks. Authors delicately designed several baselines to compare with since there's no direct comparable existing works.
* In the supplementary materials, authors verified the choice of physical constraints by ablation studies.

### Weakness:
* In my understanding, the PBD system is not learnable so the visual forces only corrects the past predictions but will not help future predictions in similar cases. In other words, this system is just trying to sync with the real-world visual perceptions but not actually "learning" to predict it. In the case when we do not have access to visual data (e.g., serving as a simulator or world model), this system will not perform better than the PBD system itself. This is not a "world model" in my opinion, it would not perform better than pure physical simulator if used as "world model".

**Quality Of The Limitations Section:**

2

**Questions For Rebuttal:**

* Does the number of Gaussians always more than particles? Is there any insights behind it? For a large Gaussian, will it be better to associate multiple particles to it?

**Robotics Focus:**

4

**Summary Of Paper:**

This paper proposed a particle-based physically enabled Gaussian Splatting approach to integrate the predictive physical modeling and visual corrections.

**Summary Of Recommendation:**

If the prediction is not learnable, I strongly doubt the contribution and impact of this work.

---

### Official Review · Reviewer_pZSN · 2024-07-23
**Very interesting work in an important direction**

**Originality:** 4
**Technical Quality:** 4
**Clarity Of Presentation:** 3
**Potential Impact:** 4
**Recommendation:** 4
**Confidence:** 5

**Review:**

- The concept of an interactive scene representation is crucial for robotics, particularly in manipulation tasks. The core idea of updating particles using Gaussian splats is both novel and logical. The authors have incorporated extensive physical priors, such as gravity, ground, and collision constraints, enhancing the realism of the simulation.

- While the core methodology is clear, some details and notations are missing or unclear, particularly in the preliminaries section.

-  The visualizations provided are excellent and effectively explain the method.

- Although a comparison with Dynamic 3D Gaussians has been made, it would be interesting to compare this approach to other classical particle-based representations that attempt to localize object particles using 3D feature matching or similar techniques.

- It would be beneficial for future comparisons if the authors released the tracking data and object meshes.

- This paper has the potential to initiate a new direction for scene representations in robotics.

**Quality Of The Limitations Section:**

3

**Questions For Rebuttal:**

- How sensitive is the method to the type of shape-matching algorithm used? Different algorithms may impact the system's performance, and insights into this aspect would be valuable.

- The correction step requires clarification. It's not immediately apparent why the Gaussians aren't moved with the particles assigned accordingly instead of the particles moving due to the visual loss. The authors should explain their reasoning behind this choice and its implications.

**Robotics Focus:**

4

**Summary Of Paper:**

The authors propose an innovative interactive scene representation for robotic manipulation using particles and Gaussian splatting. This approach employs Gaussian splatting to correct particle positions in the Position Based Dynamics (PBD) simulation while integrating other PBD priors to appropriately model robot-object interactions using particles. The method shows promising results for 3D tracking and has numerous potential downstream applications.

**Summary Of Recommendation:**

Tthis research presents an innovative approach to interactive scene representation for robotic manipulation, with the potential to significantly impact the field. There are details and clarifications missing in the current manuscript, especially in the preliminaries section, which the authors should address.

---

### Author Rebuttal · Authors · 2024-08-13

We appreciate the reviewers' valuable feedback and the time invested in reviewing our work. Attached is a zip file containing the revised paper and supplementary.

---

### Decision · Program_Chairs · 2024-09-04

**Decision:**

Accept

**Comment:**

The following strengths and weaknesses summarize the reviewers' initial comments for this submission.

Strengths:
- The proposed approach is novel and interesting.
- The paper is mostly well written and clear.
- Experimental evaluation of 2D and 3D tracking performance demonstrate better performance than provided baselines.
- An implementation with real-time tracking performance is demonstrated.

Weaknesses:
- The learning aspect of this paper is questionable, as the PBD simulator is not learnable/adapted.
- The paper lacks ablations for understanding design choices.
- The claimed usability as a world model is not substantiated in experiments.

The author response and discussion phase well addressed the reviewer concerns. The approach is mainly dynamic 3D reconstruction and updating a PBD simulator to follow the dynamic reconstruction. The learning aspect of the approach is minor as it only enables future predictions in the observed scene and the approach does not adapt properties of the dynamics models (e.g. physics parameters). Nevertheless, the reviewers find the approach relevant and interesting for the CoRL community, hence, recommending acceptance of the paper.

The revised paper must not violate the page limit for submission.